# Multisensor Magnetic Scanning Microscope for Remanent Magnetic Field Measurements

**DOI:** 10.3390/s24072294

**Published:** 2024-04-04

**Authors:** João F. Chaves, Leosdan F. Noris, Elder Yokoyama, Fredy G. Osorio G., Leonardo A. F. Mendoza, Jefferson F. D. F. Araujo

**Affiliations:** 1Department of Physics, Pontifical Catholic University of Rio de Janeiro, Rio de Janeiro 22451-900, Brazil; jhonyfelipe95@gmail.com (J.F.C.); leosdanfnoris@gmail.com (L.F.N.); ing.fredyosorio@gmail.com (F.G.O.G.); 2Department of Geosciences, University of Brasília, Brasília 70910-900, Brazil; eyokoyama@unb.br; 3Department of Electrical Engineering, State University of Rio de Janeiro-UERJ, Rio de Janeiro 20550-900, Brazil; mendonza@ele.puc-rio.br

**Keywords:** Magnetic Scanning Microscopy, Delta-Mode, remanent fields, Hall sensor, Magnetoresistance sensor

## Abstract

Magnetic Scanning Microscopy (MSM) emerged with the aim of allowing the visualization of magnetic fields of a sample or material through scanning and proved particularly useful for geology, biomedicine, characterization of magnetic materials, and in the steel industry. In this regard, the reading system of an MSM was modified using a μ-metal magnetic shielding structure to analyze remanent fields. The MSM was adapted to perform readings using two different types of sensors. The sensitive area of the sensors was evaluated, and the HQ-0811 (AKM—Asahi KaseiTM Microdevices) and STJ-010 (Micro MagneticsTM) sensors were chosen, with the HQ-0811 standardized on Printed Circuit Boards (PCBs) to facilitate handling and increase the system’s robustness. In the shielded chamber, two piezoelectric ANC-150 stepper motors (Attocube Systems) were used, arranged planarly, to allow the movement of the analyzed samples under the mounted sensors. To acquire data from the sensors, the Precision Current Source Model 6220 and the Nanovoltmeter Model 2182A (both from Keithley) were used, along with Keithley’s Delta-Mode integrated system. To analyze the system’s effectiveness, three distinct samples were analyzed for calibration, and a MATLAB program was written to analyze the images and extract the material’s magnetization. Additionally, a rock sample from the Parnaíba Basin was mapped to demonstrate the system’s capabilities.

## 1. Introduction

With the development of microscopy and magnetic sensing [1], a niche within magnetic microscopy emerged, requiring the extraction of diverse magnetic information from extensive samples [2,3,4]. To address this need, the use of spatial actuators aiding in point-to-point data collection gave rise to Magnetic Scanning Microscopy (MSM) [5,6,7,8,9,10,11]. Various operational techniques have been developed for the fabrication of MSM to measure magnetic fields and magnetization across a wide range of temporal and spatial scales [9,12,13,14,15]. Additionally, factors such as map size, sensor capability for detecting weak fields, and data acquisition time are crucial aspects in the selection of the detection technique. The final choice depends on the specific information that needs to be obtained. Following the collection of field data, the acquired information can be processed and compared with robust theoretical models developed on different platforms [10,16,17].

There is no single technique or device that suits all experiments; therefore, the choice is not trivial and depends on various factors related to the type of measurement. These factors may include the sample size, weight or quantity, device sensitivity and spatial resolution, operation at different temperatures, cost, device performance, physical state of the sample, or measurement speed [10,16,17,18,19,20]. Examples of devices that adapt to different conditions include the Vibrating Sample Magnetometer (VSM), Superconducting Quantum Interference Device (SQUID) sensors, sensors based on the Hall and Kerr effects, the inductive coil magnetometer, alternating gradient magnetometer, Magnetoresistance (MR), Giant Magnetoresistance (GMR), and Magnetic Field Camera sensors [17,20,21,22,23,24]. In general, a sample moves horizontally under a small magnetic sensor located very close to the sample’s surface. Subsequently, a magnetic map is constructed by recording the magnetic field measurement at each point in a regular grid of positions [9,13,14,15,25,26,27,28,29,30]. 

In this study, Hall effect sensors were employed to indirectly obtain information about the magnitude of a magnetic field in a specific region of space [31], rendering the effect applicable in magnetic sensors [32,33]. Additionally, the setup included Magneto-Resistive (MR) sensors, wherein the operational principle involves the alteration of material resistance when subjected to a magnetic field [34]. The more advanced versions of these sensors are based on Spin Tunnel Junctions (STJs), with the STJ model being utilized in this work [35,36,37]. After preliminary tests with known samples, a rock sample from the Parnaíba Basin was mapped to showcase the capabilities of this integrated system.

## 2. Magnetic Scanning Microscopy Layout

Within the chamber’s interior, an acrylic base is situated to accommodate the nanopositioners, along with a digital camera providing real-time visualization during measurements (which can be deactivated if necessary). Photographs depicting the internal assembly of the chamber can be observed in Figure 1a, which showcases the support structure for sample holders, piezoelectric elements, and the digital camera; Figure 1b displays the HQ-0811 sensor (Manufacturer Asahi Kasei^TM^ Microdevices, Tokyo, Japan); and Figure 1c features the STJ-010 sensor (Manufacturer Micro Magnetics^TM^, Fall River, MA, USA).

The physical structure of the MSM consists of a four-layer μ-metal cylindrical chamber (a Ni-Fe ferromagnetic alloy typically employed in magnetic shielding) that serves as a shield against external magnetic fields, particularly those of low frequency. Said chamber has a height of about 191 cm, having four layers of about 1.5 mm thickness, distributed in doubles. The external double layer diameter is about 70 cm, while the internal double layer chamber is about 40 cm. Also, it should be noted that, when opened, the internal chamber access is a cylindrical window of about 18 cm in height with a circumference arc length of 19.5 cm. Within the shielded chamber, two planarly arranged model ANC-150 piezoelectric step motors (Manufacturer Attocube Systems, Haar, Germany) [38,39] were employed to facilitate the movement of analyzed samples beneath the mounted sensor. This structure was constructed by Eduardo A. Lima and his collaborators [8].

### 2.1. Sensors and Response Curves (RC)

One of the crucial aspects in assembling the MSM is selecting an appropriate sensor with essential qualities for the intended measurements. To achieve this, an analysis was conducted to assess the characteristics of the chosen sensors for evaluation.

One of the first characteristics that can be looked at for choosing this is integration capability for the already built structure. Sensors like the SQUID need a cryogenic supporting apparatus that would involve major changes to the structure. Meanwhile, sensor systems commonly used in Magnetometers use applied fields to make measurements, which is not adequate for remanent field readings.

The options are then reduced to Magnetoresistance and Hall effect-based sensors. These sensors were then evaluated by using the available datasheets and checking signal-to-noise ratios. The following outlines the methodology employed.

#### 2.1.1. Resolution

For sensors relying on the Hall effect to extract information about the remanent magnetic field (micromagnetism), it is crucial that the scannable area be minimized. In other words, measurements must be taken in a space equivalent to the size of the sensor’s sensitive area, and that should be minimal. Thus, the HQ-0811 sensor from Asahi Kasei^TM^ Microdevices brand [40], with a sensitive area of 200 μm, was chosen for use in the MSM, which was identified as the Hall Magnetic Scanning Microscope (HMSM). In addition to this, the system was also tested with the STJ-010 Magnetoresistance sensor from Micro Magnetics^TM^ brand [36], which was identified as the Magnetoresistance Magnetic Scanning Microscope (MMSM).

#### 2.1.2. Printed Circuit Boards

After selecting the sensors to be used, during the initial stages of the research, it was decided that the HQ-0811 sensors would be standardized through circuit boards. This decision aimed to streamline handling and strengthen connections during assembly. A specific project was developed using the Ki-Cad program for the production of these boards, as illustrated in Figure 2. The local company responsible for the manufacturing and testing of these boards was PRV TECH (Manufacturer Soluções Eletrônicas, São Paulo, Brazil). Following the acquisition of the boards, DC field tests were conducted, generated by a pair of Helmholtz coils, with the respective collection of RCs.

#### 2.1.3. Signal-to-Noise Ratio

A second important characteristic to be studied is how large the signal measured by the sensor (in the case of Hall sensors, the transverse potential) is compared to the background noise. Due to the distinct operating method of magnetoresistance sensors, where both the power supply and the reading are conducted in parallel, the STJ-010 sensor was not evaluated in the manner described below.

The specifics for the evaluation are as follows: A custom coil, with previously calculated and tested field/current ratio (7.7 × 10^−4^ T/A in SI units), was assembled on an acrylic support. The HQ-0811 sensor to be tested was positioned where the highest field from the RC was measured. The field intensity in this position was measured using a model Series 9950 Gauss/Teslameter from the F.W. Bell brand. The sensor was supplied with a current of 3 mA in the same configuration to be used in future measurements. The coil was powered with an alternating square wave current of 105 mA in two distinct frequency configurations, 4 Hz and 80 Hz (See Figure 3).

A Stanford Research Systems^TM^ (Sunnyvale, CA, USA) model SR760 FFT spectrum analyzer, controlled by a LabVIEW acquisition and control program, receives the transverse potential output from the HQ-0811 sensor, generating a frequency by signal intensity graph. Evaluations are performed in four distinct configurations: field frequencies of 4 and 80 Hz with the μ-metal chamber open (see Figure 3a,b); field frequencies of 4 and 80 Hz with the μ-metal chamber closed (Figure 3c,d). These configurations cover the sensor’s operation in both high and low frequencies and will serve as a comparison regarding the effectiveness of the magnetic shielding of the chamber. The signal-to-noise ratio is calculated by taking the maximum signal peak found on the graph and subtracting it from the background noise average.

It should be noted that in both situations, the signal-to-noise ratios obtained with the closed chamber are consistently better, with improvements ranging from 10% to 25% compared to the open chamber. The initial proposal for the application of such a chamber was to isolate the experiments contained within it from background noise, especially low-frequency noise (given that the experiments conducted within it also occur under low-frequency conditions). The data in question demonstrate that the magnetic shielding appears to be effective in improving signal acquisition.

### 2.2. Delta-Mode Acquisition

The Delta-Mode^TM^ is a measurement acquisition mode developed by Keithley, where data collected during current reversal is processed by averaging, with the aim of eliminating Thermal Electromotive Force (thermal EMF) noise that may affect the obtained data. The mode operates through conjunction of a Precision Current Source Model 6220 and a Nanovoltmeter Model 2182A both of the Keithley brand, and details about its operation can be found in the company manuals [41,42].

The method first applies a positive current, resulting in voltage T1; then, the polarity is reversed, giving voltage T2. Finally, the average of these is calculated, resulting in the actual measurement. A practical example would be a measurement in an ohmic circuit, where the resistance is 10 kΩ with a delta current of 1 mA. Disregarding thermal effects, the expected voltage in the circuit would be 10 V for T1 and −10 V for T2. However, if thermal effects introduce a noise of 10 mV, the values of T1 and T2 would be altered, giving 10.1 V and −9.9 V, respectively. After averaging, the final value would then be 10 V, thus returning the actual measurement without thermal EMFs.

The Keithley instruments used in this case, namely the Precision Current Source Model 6220 and Nanovoltimeter Model 2182A [41,42], employ a moving measurement window for averaging: each Delta acquisition cycle involves 3 changes in the polarity of the applied current, with a weighted average where the intermediate measurement has double the weight. Each measurement is also multiplied by a negative term with an order dependent on the number of cycles, ensuring that the measurements are always positive.

### 2.3. LabVIEW Automation

With the aim of providing the necessary communication between sensors (Hall Effect and STJ), piezoelectric actuators, and the computer, a LabVIEW^®^ version 2017 routine was modified and adapted from previous works [8]. This program aims to keep all input and output controls of the MSM in a single platform as well as to standardize the reading in text files that will be subsequently read by auxiliary analysis programs in MATLAB ^®^ version R2023a. The control panel can be viewed in Figure 4a.

As shown in Figure 4b (blue box on Figure 4a), it is possible to control the inputs (power to connected elements as well as which sensors are used), compliance (voltage limit to prevent damage to powered elements), and allow offset adjustments (zeroing the measurement). Similarly, the program gives options over scanning characteristics, like the possibility to customize the spatial resolution of the map to be created (control over step size), as seen in Figure 4c (red box on Figure 4a). Finally, there is both a signal-by-time graph (bottom center on Figure 4a), to keep track of readings, and a real-time preview of the magnetic field map (right side on Figure 4a).

## 3. Test Analysis

For an evaluation of the MSM’s functionality, three calibration samples were chosen: a sample holder with two wells approximately 200 μm in radius and 100 μm in depth, filled with iron oxide powder, or “rust”, and identified as double iron oxide, shown in Figure 5a; a sample holder with three wells arranged in a triangular pattern, each with a radius of 200 μm and a depth of 400 μm, separated by about 900 μm when closer and 1300 μm when farther apart, filled with magnetite, and identified as triple magnetite, shown in Figure 5b; and a sample holder filled with a nickel wire, with dimensions of 125 μm in radius and 480 μm in height and identified as nickel sample, shown in Figure 5c. Additionally, to test the capabilities of the MSM, a study was conducted on a rock from the Jurassic–Triassic basaltic dike cluster, which was extracted from the western edge of the Parnaíba Basin and identified as Rock Sample, presented in Figure 5d.

Regarding the samples, relevant information to be provided is as follows: the double iron sample was prepared using iron oxide powder, identified as Supermagna RW 222 from the company Metal-Chek, assumed to be 96% magnetite; the triple magnetite sample consisted of magnetic microparticles of 99% magnetite; the nickel sample was a nickel wire with 99.9% purity, previously used in other works for calibration [11]. All samples were used for test measurements, with analysis by the Mapper program (Developed for this study). The obtained results were compared with measurements made by an MSM owned by the research group where this work was developed, previously reported in other works.

### 3.1. Data Treatment

After data acquisition it is crucial that, before the analysis process, the data is processed, both to eliminate any background noise and smooth the whole. For this purpose, a program in MATLAB, identified as Mapper, was developed. The Mapper program has been previously used and described in [11]. The sample to be used in the test case was set up, consisting of a nickel wire with 99.9% purity. The sensors used in this case were the Hall-0811 from the Asahi Kasei™ brand and the STJ-010 from Micro Magnetics™ brand, and both sensors were powered using Delta-Mode™ with ±3 mA and ±5 µA, respectively [41].

#### 3.1.1. Configuration and Data Acquisition

By utilizing data acquired from the LabVIEW program, the Mapper program generates a surface plot where the X and Y axes represent measurement positions, and the Z axis represents the voltage values obtained by the sensor at those positions. Consequently, with a prior calibration of the sensor’s response to magnetic fields, a conversion can be applied to transform the units on the Z axis from Volt to Tesla.

Geometric characteristics of the sample holders (depth and diameter of wells) and the average density of the material placed in them are provided to the program. After building the magnetic field maps from the readings and performing some image filtering, the treated maps can be visualized in Figure 6. For data processing, the algorithm initially scans the original map in search of the absolute maximum value, which will be used as a reference. Then, a differentiation along the X axis is performed across the entire map, generating the normalized gradient map, as shown in Figure 6b for the HQ-0811 sensor and Figure 6d for the STJ-010 sensor. It can be noted that the map in Figure 6c has better resolution than the map in Figure 6a. The magnetic field intensity of the STJ-010 sensor is lower than that of the HQ-0811 sensor, possibly due to the manual positioning of the STJ-010 sensor along the Z-axis (distance between the sensor and the sample).

#### 3.1.2. Fits and Magnetization

After image processing, the algorithm performs a fitting process to obtain the magnetization of the sample. The fitting is performed accounting for the info first given about the sample, and with that the sample magnetization is obtained by comparing the acquired data with a model of a current cylinder (the same shape as the sample holder), calculated by Biot–Savart’s Law. In this calculation, the dimensions and material density are used to estimate the mass of the cylinder, resulting in a current density ratio in the cylinder’s area, in Am²/kg. Also, it is advised to give an initial value for the magnetic moment of the sample (it is useful to approximate this initial value to an expected value, saving data processing time and expediting the fitting process) [7].

The fit in question is a current cylinder model using Biot–Savart, presented in Equation (1):(1)$$B\left(x,y,z\right)=\frac{\mu_{0}}{4\pi }\int_{circuit}\frac{Idl\times r^{\prime }}{{r^{\prime }}^{3}}$$
where μ_0_ is the magnetic permeability of the vacuum, I is the current in the cylinder, and r′ is the position vector of the field measurement point relative to the cylinder. Developing this term appropriately for the geometry of the system in question, we obtain the following:(2)$$\begin{array}{l} B_{z}\left(x^{\prime },y^{\prime },z^{\prime }\right)\\ =\frac{\mu_{0}I}{\pi l}\int_{0}^{2\pi }\frac{R(R\;-\;xcos\left(\emptyset \right))}{{\left[{\left(x\;-\;Rcos\left(\emptyset \right)\right)}^{2}+{\left(Rsin\left(\emptyset \right)\right)}^{2}+{\left(Sensordistance\;-\;h\right)}^{2}\right]}^{3/2}}d\emptyset \end{array}$$
where R is the radius of the cylindrical cavity, with x modeling one of the axes in the XY plane, l is the height of the cylindrical cavity, ∅ is the angle in the XY plane, Sensordistance is literally the distance from the sensor to the top of the sample, and h is the variable modeling the axis perpendicular to the XY plane.

The fitting process can be divided into three parts, where initially, the algorithm searches for the line containing the highest absolute peak in the map and creates an analysis window along this line. After that, the “initial value” of magnetic moment is used to attempt to model the sensor distance, followed by the same process to model the magnetic moment, and finally, the magnetization. That process is looped, and in each cycle, the value of the sought parameter is altered and the mean squared error between the model and experimentally obtained values in a window near the peak of the curve is calculated. Upon reaching the value that minimizes the mean squared error, the algorithm stores the parameter value as definitive. It is important to note that the comparative process is conducted only in a small window near the maximum value found in the map; the aim is to mitigate the adverse effects that smoothing may have on the model. In the end, results were obtained for the remanent magnetization and distance of 4.25 Am^2^/kg and 160 µm, respectively, for the HQ-0811 sensor. The distance obtained from the model for the STJ-010 sensor was 147 μm, and the magnetization was 2.42 Am^2^/kg.

Figure 7 presents a comparison of the theoretical fit with the experimental results, where M represents magnetization, DS is the sensor distance, MSE is the mean square error, and EPP is the percent peak error. One noticeable characteristic in the fit is how broad the signal appears compared to the model. Among the supposed reasons for this is the fact that the sensor’s resolution allows signal “leakage” from nearby sites, the substantial difference between the resolution of the sensor (200 μm) and the spatial resolution of the positioners (30 μm), and the smoothing process performed during the image treatment, which reduces signal gradients, broadening measurements and softening/reducing peaks.

## 4. Results

For the double iron oxide sample, one of the initial points to be considered is that the sample holder has a small rust powder spot resulting from its preparation, and its detection is not clear on the HMSM due to the observed noise. This spot is noticeable in the photo of the double iron sample in Figure 5a, just below the two filled wells. Figure 8a presents the treated map for the double iron sample measured with the HQ-0811 sensor, and in Figure 8c, the measurement with the STJ-010 sensor is shown. In addition, gradient maps are displayed in Figure 8b and Figure 8d, respectively.

It is possible to observe that the treatment manages to distinguish the signal from the two wells, although there is still signal overlap between them. Another point to consider is that the sensor was not able to differentiate the previously mentioned stain from the background noise in the case of the HQ-0811 sensor. In the case of the STJ-010 sensor, it can indeed distinguish the stain, and this fact is much more evident in the gradient map (Figure 8d). Using the map from Figure 8a, the average height of the sensor relative to the top of the sample holder was found to be about 149 μm, which is consistent with the expected layout (measurements for the HMSM are always taken with the sensor “touching” the sample). Additionally, a remanent magnetization of approximately 6.40 Am^2^/kg was found, with a percent error between the theoretical and experimental peaks of about 10%.

In the map from Figure 8c, it is noticeable that the resolution of the STJ-010 seems to be much better and less noisy than that obtained with the HQ-0811 sensor. Moreover, as evident, the STJ-010 was able to capture the signal generated by the rust stain on the sample holder, discerning it from the background noise. From the fittings, it was extracted that the average distance of the sensitive area of the STJ-010 sensor is approximately 317 µm from the sample, and the magnetization obtained from the map is about 6.39 Am^2^/kg, which closely matches the results obtained by the HQ-0811 sensor for the double iron sample. The relative errors between the peak data and the peak of the fit were calculated, taking the ratio between the values and resulting in 5.9% and 5.4%, respectively.

For the triple magnetite sample, the treated and filtered maps are presented in Figure 9a using the HQ-0811 sensor and Figure 9c using the STJ-010 sensor. Additionally, the gradient maps are shown in Figure 9b and Figure 9d, respectively. As observed, the data treatment with the HQ-0811 sensor was not entirely effective; the high noise present in the original map was reduced but not completely eliminated.

In this case, the distance from the sensor and the magnetization obtained from the data treatment are 107 μm and 6.25 Am^2^/kg, respectively, for the HQ-0811 sensor. The magnetization obtained by the STJ-010 sensor was approximately 6.41 Am^2^/kg, with a distance of about 292 μm.

As a matter of comparison for the results achieved, Table 1 presents the obtained Mass Magnetizations (MM) and the Percentage Errors (PE) between them for each used sample.

For samples based on magnetite, namely double iron oxide and triple magnetite, the magnetizations obtained have errors in the range of 1% to 3% for both sensors. It can be concluded that the equipment was successful in extracting this information from the samples through the treatments. However, due to the lack of an exact measurement for the magnetization of the nickel sample, it is valid to assume that the comparison made may not be correct. One of the difficulties when working with a Ni wire is the anisotropic nature of its shape, which can give rise to differences in the magnetization values depending on how it is oriented in relation to the sensing area. It can be pointed out then that the shown difference reported in the results can be due to these anisotropies. A notable point to “support” the values obtained by the HMSM and MMSM is the similarity between the values obtained, even though they are from different sensor systems. When compared to each other, taking the lower of the values (closer to the true value), the MMs differ by about 3%.

## 5. Rock Sample Analysis

For the analysis of the rock sample, the sample was divided into three regions, since the dimensions of the sample exceed the ranges of the stepper motors. The regions mentioned are shown in Figure 10.

Furthermore, for comparison purposes and better orientation, the RS was also measured using another auxiliary system based on HQ-0811 sensors, as shown in Figure 11 (image in the center). The map in question, which we will identify as the control map, capable of covering the entire area of the rock, was obtained using equipment whose results have already been evaluated by the scientific community, as observed in works [10] and [11].

Within these three large areas, four regions that can be identified at a glance were identified. In the case of the Hall sensor (HQ-0811), the measurements were taken as indicated in Figure 12.

As can be noticed in Figure 11 and Figure 12, the map taken from the auxiliary equipment shows some characteristics of the rock sample, like high intensity magnetic islands and the general magnetic profile of it. However, it needs to be acknowledged that the auxiliary equipment needs an applied field to measure such profiles. By using the MSM, in both modes, the magnetic shielding allows for the measurement of very fine details of said islands without any external magnetic fields. Such a change allows for the imaging of remanent magnetic fields that would otherwise lack resolution if made out of a shielded chamber.

## 6. Conclusions

This work was conceived as an adaptation of a structure previously owned by the laboratory in the construction of an MSM, which would focus on measuring remanent magnetic fields with high spatial resolution using a new reading system for this structure and making the system adaptable to different sensors. The developed system exhibits appreciable simplicity when compared to previous setups and is compatible with sensors that fit its power and acquisition systems, which prove it to be a versatile apparatus. It has been tested using an HQ-0811 Hall effect sensor and an STJ-010 Magnetoresistance sensor. To assess the equipment’s capabilities, characterizations of the equipment were performed, including signal-to-noise ratios, sensor calibrations, and tests with synthetic samples and rock samples. For data processing, the Mapper program was developed in MatLab, which allowed for improved maps and subsequent extraction of magnetic information such as the magnetization of the mapped sample.

## Figures and Tables

**Figure 1 sensors-24-02294-f001:**
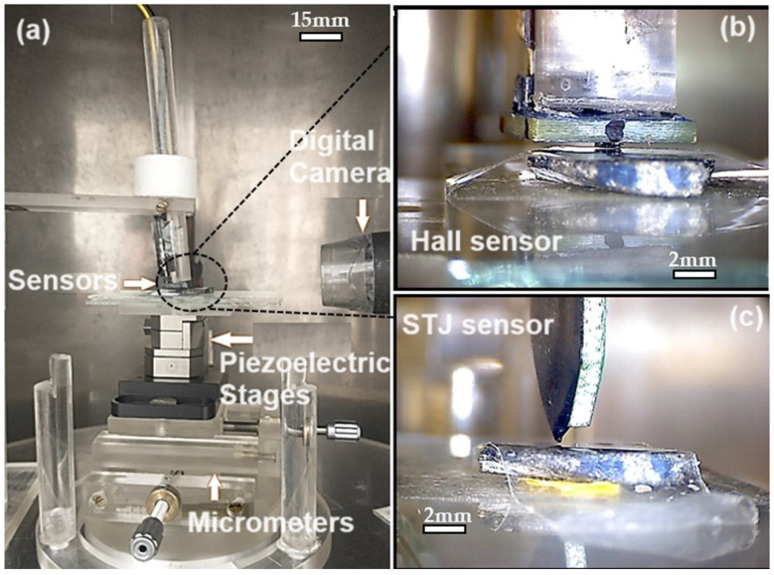
(**a**) Layout of the MSM with the HQ-0811 sensor, identifying its components, and the image captured by the USB camera. (**b**) HQ-0811 sensor and (**c**) STJ-010.

**Figure 2 sensors-24-02294-f002:**
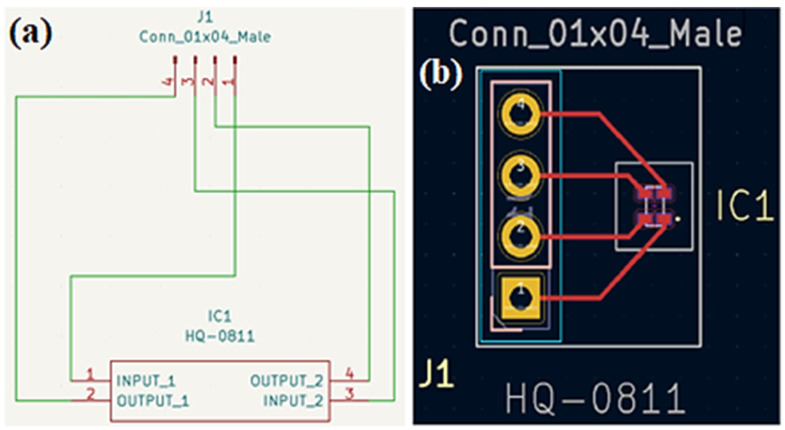
(**a**) Circuit diagram of the boards. (**b**) PCB file for the boards, created in Ki-Cad.

**Figure 3 sensors-24-02294-f003:**
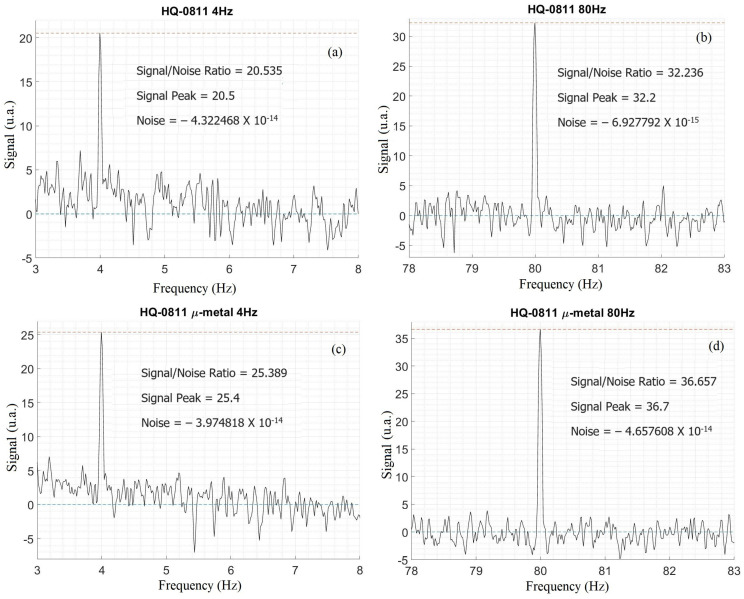
Graphs of the signal-to-noise ratio for the HQ-0811 sensor with the μ-metal chamber open, (**a**) 4 Hz and (**b**) 80 Hz; with the μ-metal chamber closed, (**c**) 4 Hz and (**d**) 80 Hz.

**Figure 4 sensors-24-02294-f004:**
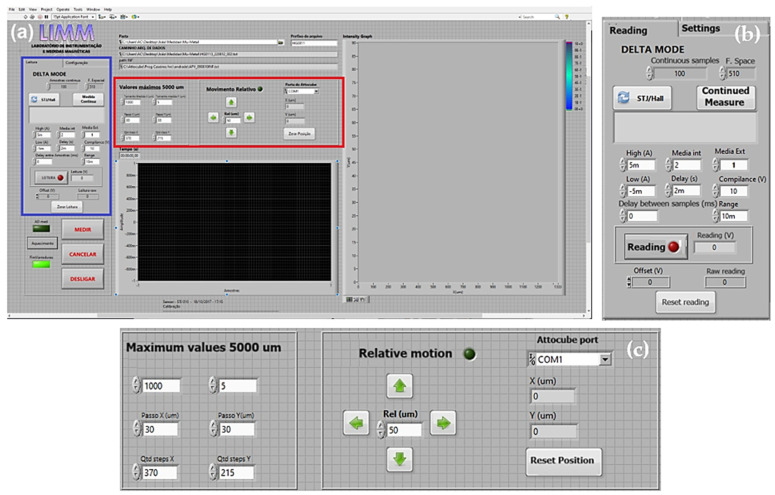
(**a**) Control panel of the communication program, (**b**) Delta-Mode control responsible for power and reading configurations, and (**c**) step control and scanning area.

**Figure 5 sensors-24-02294-f005:**
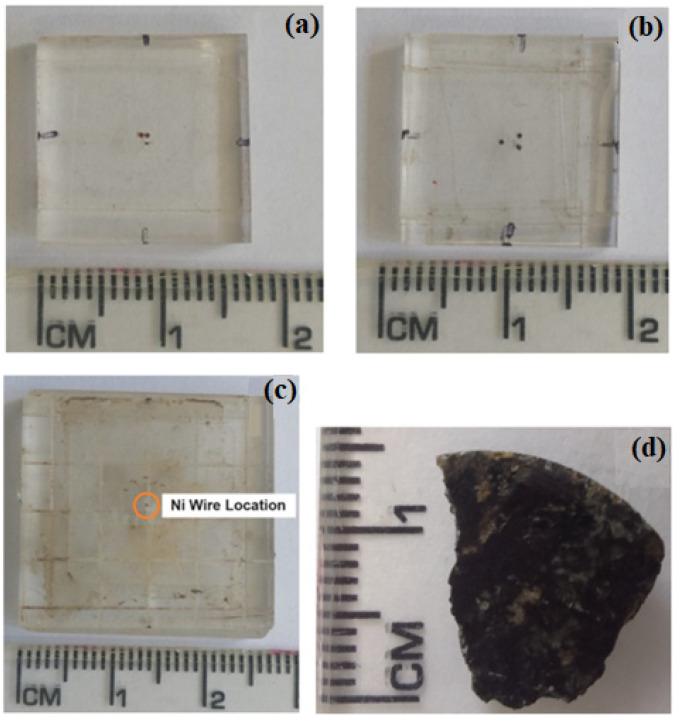
Sample holder with ruler, featuring (**a**) double iron oxide, (**b**) triple magnetite, (**c**) nickel sample, and (**d**) rock sample.

**Figure 6 sensors-24-02294-f006:**
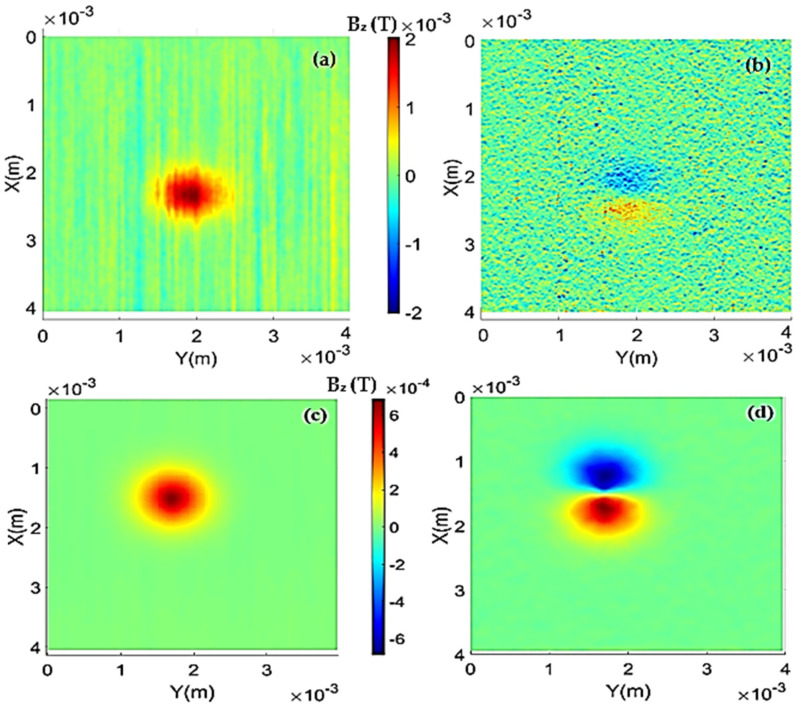
(**a**) Treated and filtered map of the nickel wire using the HQ-0811 sensor. (**b**) Normalized gradient map using the HQ-0811 sensor. (**c**) Treated and filtered map of the nickel wire using the STJ-010 sensor. (**d**) Normalized gradient map using the STJ-010 sensor.

**Figure 7 sensors-24-02294-f007:**
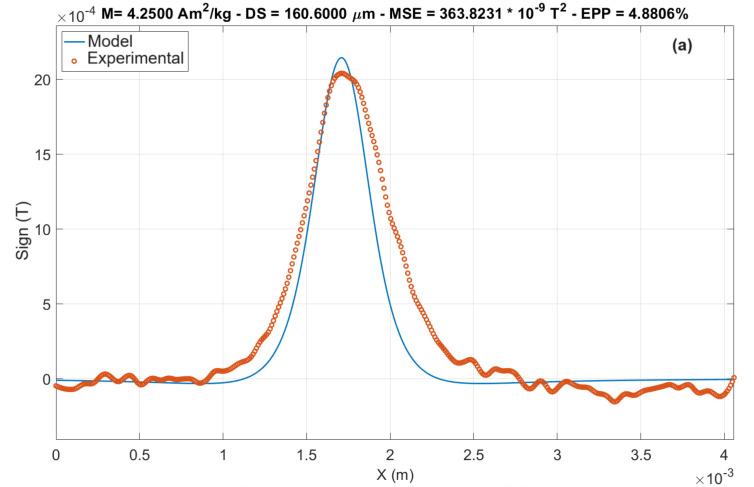
Attempted fit of the nickel wire, adjusted to the experimental data, in the window around the peak; (**a**) HQ-0811 sensor and (**b**) STJ-010 sensor.

**Figure 8 sensors-24-02294-f008:**
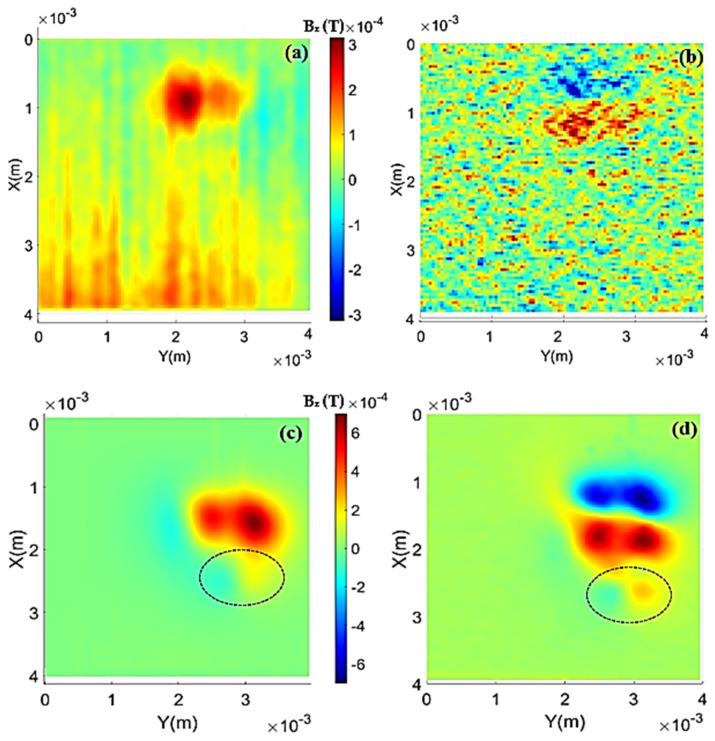
(**a**) Treated map of the double iron oxide sample obtained with the HQ-0811 sensor. (**b**) Normalized gradient map obtained with the HQ-0811 sensor. (**c**) Treated map of the double iron oxide sample obtained with the STJ-010 sensor. (**d**) Normalized gradient map obtained with the STJ-010 sensor.

**Figure 9 sensors-24-02294-f009:**
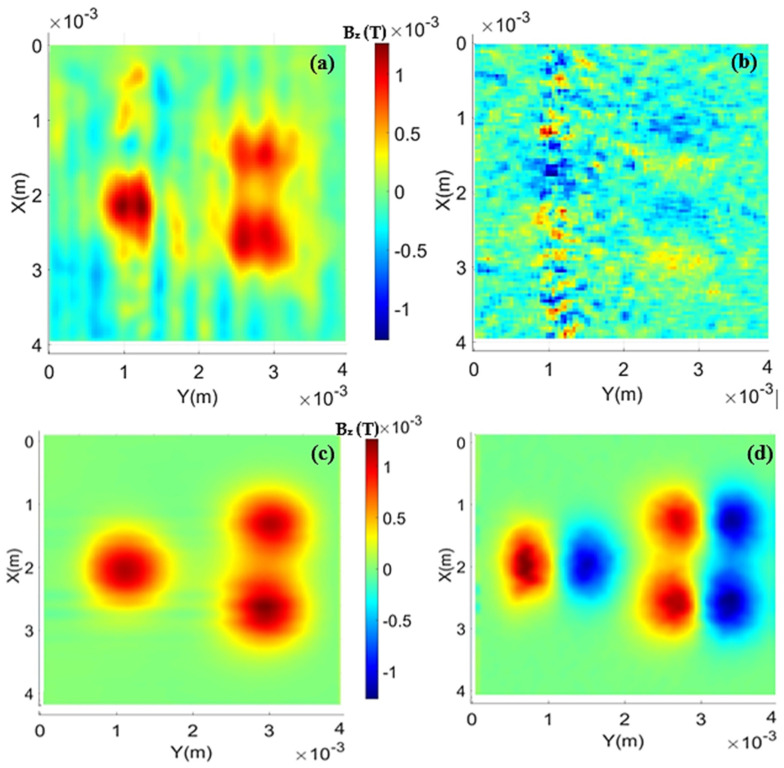
Treated maps for the triple magnetite sample: (**a**) for the HQ-0811 sensor and (**c**) STJ-010 sensor. Normalized gradient maps: (**b**) HQ-0811 sensor and (**d**) STJ-010 sensor.

**Figure 10 sensors-24-02294-f010:**
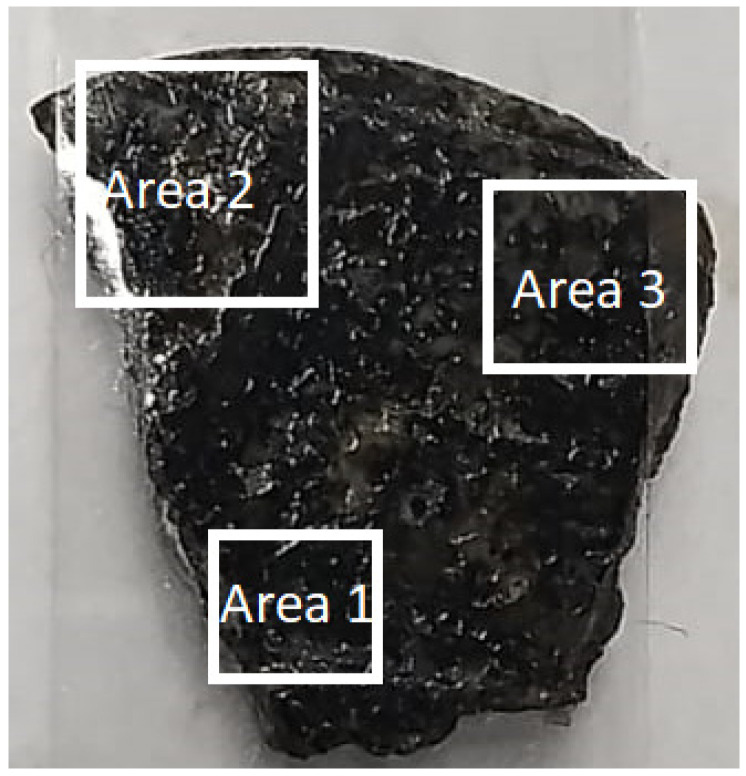
Approximate areas analyzed in the RS.

**Figure 11 sensors-24-02294-f011:**
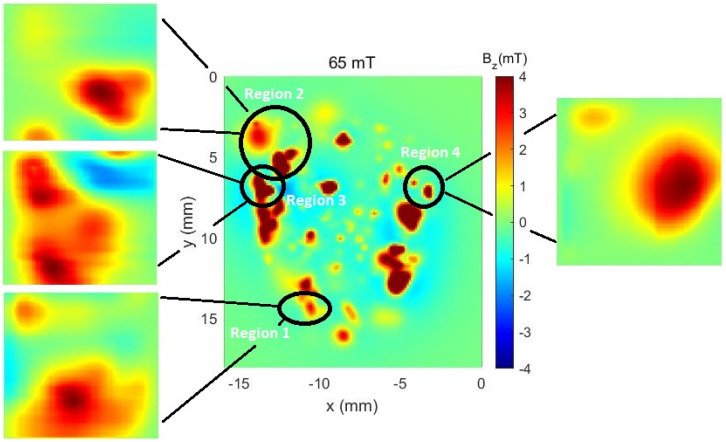
The control map, obtained from auxiliary equipment (center of the image), along with the maps obtained with the STJ-010 sensor identified for each of the measured regions.

**Figure 12 sensors-24-02294-f012:**
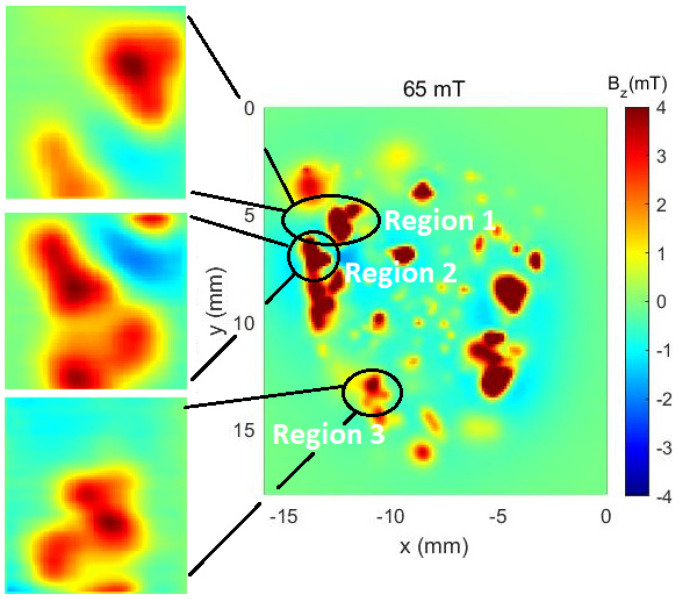
The control map obtained from the auxiliary equipment (right image). Additionally, the maps obtained with the HQ-0811 sensor identified for each of the measured regions are shown.

**Table 1 sensors-24-02294-t001:** Comparative table of Mass Magnetizations obtained for each analyzed sample, along with their respective errors.

Sample	MM (Am^2^/kg)	PE (%)
Double Iron (HQ-0811)	6.40	0.16
Double Iron (STJ-010)	6.39	
Triple Magnetite (HQ-0811)	6.25	2.56
Triple Magnetite (STJ-010)	6.41	
Nickel Sample (HQ-0811)	4.25	43.06
Nickel Sample (STJ-010)	2.42	

## Data Availability

Data are contained within the article.

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
