# Peer review of "Multisensor Magnetic Scanning Microscope for Remanent Magnetic Field Measurements"

_sensors, 2024, doi:10.3390/s24072294_

Round 1

Reviewer 1 Report

Comments and Suggestions for Authors

The authors have introduced a multisensor MSM for remanent magnetic field measurements based on the HQ-0811 and STJ-010 sensors. While I applaud the author’s efforts, I hold the view that the manuscript, in its present form, may not meet the necessary criteria for publication. Below are the main reasons for my decision.

1. The MSMs reported in this study are constructed using commercially available sensors and data acquisition instruments, and a previously reported data treatment based on MATLAB. The novelty of the MSM system and its superior performance need clarification. Authors should elucidate the unique aspects of the MSM system and demonstrate its enhanced performance compared to similar systems reported previously.

2. In the introduction part, the authors have listed various types of magnetic sensors such as Vibrating Sample Magnetometer, Superconducting Quantum Interference Device sensors, sensors based on the Hall and Kerr effects, inductive coil magnetometers, alternating gradient magnetometers, and Magnetoresistance  and Giant Magnetoresistance sensors. The authors should provide more detailed explanations regarding the advantages and disadvantages of these sensor types. Additionally, they should justify their selection of Hall effect sensors and STJ sensors for this study.

3. Figure 6 and Figure 7 illustrate significant discrepancies in the test results for Nickel Wire obtained using the HQ-0811 Sensor and STJ-010 Sensor, showing a substantial difference of approximately 3-fold. These inconsistencies raise doubts regarding the reliability of the MSM's test results. Authors should address this issue and provide explanations for the large variations observed between the two sensors.

Regrettably, considering the aforementioned points, I cannot recommend accepting this paper for publication in its current state.

Comments on the Quality of English Language

Minor editing of English language required.

Author Response

Author's Reply to the Review Report (Reviewer 1)

Thank you very much for reviewing our paper

The authors have introduced a multisensor MSM for remanent magnetic field measurements based on the HQ-0811 and STJ-010 sensors. While I applaud the author’s efforts, I hold the view that the manuscript, in its present form, may not meet the necessary criteria for publication. Below are the main reasons for my decision.

  1. The MSMs reported in this study are constructed using commercially available sensors and data acquisition instruments, and a previously reported data treatment based on MATLAB. The novelty of the MSM system and its superior performance need clarification. Authors should elucidate the unique aspects of the MSM system and demonstrate its enhanced performance compared to similar systems reported previously.

(Lines 369 – 380) (Lines 390-397)

 It is explained how this MSM is different when compared to the previously reported auxiliary equipment.

Our system is different because it has the versatility of multisensor use. These said sensors have different magnetic behaviours, but can be used by the same acquisition structure. It’s also noted that this was the first time that a Delta Mode based reading system was used in this type of acquisition structure (especially considering the multisensory characteristic of the MSM). Furthermore, no commercial equipment similar to this multi-sensor system was found in the literature.

  1. In the introduction part, the authors have listed various types of magnetic sensors such as Vibrating Sample Magnetometer, Superconducting Quantum Interference Device sensors, sensors based on the Hall and Kerr effects, inductive coil magnetometers, alternating gradient magnetometers, and Magnetoresistance  and Giant Magnetoresistance sensors. The authors should provide more detailed explanations regarding the advantages and disadvantages of these sensor types. Additionally, they should justify their selection of Hall effect sensors and STJ sensors for this study.

Some of these sensor were better described in the text (90-97) with the reasoning on why they were not chosen. Kerr sensors are based on optical effects and need lasers and a more complex structure to be built inside the chamber, and was discarded.

It is important to remember that the article dealt with a microscope and not a magnetometer, for this reason several sensors mentioned above by the reviewer are not suitable for this type of mounted instrument.

  1. Figure 6 and Figure 7 illustrate significant discrepancies in the test results for Nickel Wire obtained using the HQ-0811 Sensor and STJ-010 Sensor, showing a substantial difference of approximately 3-fold. These inconsistencies raise doubts regarding the reliability of the MSM's test results. Authors should address this issue and provide explanations for the large variations observed between the two sensors.

The discrepancies mentioned can be see in the graph of figure 6 as being about 2 * 10-3 T for the HQ-0811 sensor and 6 * 10-4 T for the STJ-010. This difference is expected due to the different structures and robustness of the sensors, STJ sensors are generally put further away due to being more fragile in an attempt to keep its integrity. Mentioned in (line 235-237). Another aspect that can be a factor, and is also discussed in the paper, is related to the anisotropy of the wire.

Regrettably, considering the aforementioned points, I cannot recommend accepting this paper for publication in its current state.

Reviewer 2 Report

Comments and Suggestions for Authors

The authors have set up magnetic scanning microscopes to characterize remanent magnetic fields from various magnets. Their work is strengthened by the comparison of Spin Tunnel Junctions with Hall sensors. I recommend the publication of this manuscript after a minor revision.

1.    The manuscript is generally well-written, but I suggest the authors consider rewriting the final paragraph of the introduction. Avoiding experimental details in this section and providing a concise final paragraph will better engage readers for the subsequent sections.

2.    It would be beneficial for the authors to increase the number of references from recent journal publications. Including references to new magnetic instruments, such as a magnetic camera in Appl. Sci. 2021, 11, 3302, will broaden the perspectives of readers. (https://doi.org/10.3390/app11083302)

3.    Unnecessary abbreviations, such as MMVH, MMV, DI, TM, NS, and RS, should be avoided for clarity and consistency.

4.    The caption for Figure 5(d) is missing, and should be added for clarity.

5.    The subsections in the Results section should be restructured, and the maps from Nickel samples should be presented in this section for better organization.

6.    The authors should consider changing the term “Magnetization Value” to “Mass Magnetization” for improved clarity.

7.    The sentence “However, the system is still capable of imaging and image processing, as will be presented below.” is an incorrect way to end the Results section. Consider revising for clarity and coherence.

8.    The authors should proofread the manuscript to eliminate typographic errors, such as “integrated system..” on page 2. Additionally, ensure that the references are in a consistent format according to the journal's guidelines.

Comments on the Quality of English Language

There are grammatical and typographic errors. The manuscript needs language editing and prooreading.

Author Response

Author's Reply to the Review Report (Reviewer 2)

Thank you very much for reviewing our paper

The authors have set up magnetic scanning microscopes to characterize remanent magnetic fields from various magnets. Their work is strengthened by the comparison of Spin Tunnel Junctions with Hall sensors. I recommend the publication of this manuscript after a minor revision.

  1. The manuscript is generally well-written, but I suggest the authors consider rewriting the final paragraph of the introduction. Avoiding experimental details in this section and providing a concise final paragraph will better engage readers for the subsequent sections.

Agreed and changed (lines 56-64)

  1. It would be beneficial for the authors to increase the number of references from recent journal publications. Including references to new magnetic instruments, such as a magnetic camera in Appl. Sci. 2021, 11, 3302, will broaden the perspectives of readers. (https://doi.org/10.3390/app11083302)

Agreed and changed

Reference inserted. (lines 51-52) and (lines 457-458)

  1. Unnecessary abbreviations, such as MMVH, MMV, DI, TM, NS, and RS, should be avoided for clarity and consistency.

Agreed. Such abbreviations have been reduced, with the ones for the samples being discarde

  1. The caption for Figure 5(d) is missing, and should be added for clarity.

Agreed and changed (lines 209-210)

  1. The subsections in the Results section should be restructured, and the maps from Nickel samples should be presented in this section for better organization.

Partially agreed, got rid of the subsectioning but kept the Ni samples on the previous section. The reason is that we still belive that’s good to keep in mind that the Ni sample is treated more like an example for the method and instrument than a result.

  1. The authors should consider changing the term “Magnetization Value” to “Mass Magnetization” for improved clarity.

Agreed and Changed in lines (351-368)

  1. The sentence “However, the system is still capable of imaging and image processing, as will be presented below.” is an incorrect way to end the Results section. Consider revising for clarity and coherence.

Agreed and changed for better clarity (lines 390 - 397)

  1. The authors should proofread the manuscript to eliminate typographic errors, such as “integrated system..” on page 2. Additionally, ensure that the references are in a consistent format according to the journal's guidelines.

Agreed and Changed

Reviewer 3 Report

Comments and Suggestions for Authors

Manuscript ID: sensors-2924663

Title: Multisensor Magnetic Scanning Microscope for Remanent Magnetic Field Measurements

This manuscript presents a reading system of an MSM (Magnetic Scanning Microscopy) that was modified using a μ-metal magnetic shielding structure to analyze remanent fields. The MSM was adapted to perform readings using two different types of sensors. The manuscript is not well organized and needs to be improved. The reviewer has some comments below.

1.       This manuscript is like a book chapter. Revise it to be more concise by reducing the number of sub-sections and focusing on the main work.

2.       Fig. 1 is a good photo for demonstrating the outcome of this work. However, the author needs to discuss it in more detail.

3.       Compare the results of this work with other configurations.

4.       Highlight the new contribution of this work to the research field. I know this work proposes a new reading system and makes the system adaptable to different sensors. Nevertheless, the author should highlight it in Section 3 or Section 4. Some comparison or Schematic of the method may help.

5.       Considering: remove Fig. 4, font size too small to read, Fig. 5 should retake, rearrange for better quality, Fig. 6, 8, 9 move the labels (a-d) to the right corner, Fig. 7 why two labels (a), (b) are double.

6.       The English, symbols, and presentation need to be improved.

Comments on the Quality of English Language

Moderate editing of English language required

Author Response

Author's Reply to the Review Report (Reviewer 3)

Thank you very much for reviewing our paper

Title: Multisensor Magnetic Scanning Microscope for Remanent Magnetic Field Measurements

This manuscript presents a reading system of an MSM (Magnetic Scanning Microscopy) that was modified using a μ-metal magnetic shielding structure to analyze remanent fields. The MSM was adapted to perform readings using two different types of sensors. The manuscript is not well organized and needs to be improved. The reviewer has some comments below.

  1. This manuscript is like a book chapter. Revise it to be more concise by reducing the number of sub-sections and focusing on the main work.

Agreed, reduced the number of sub-sections (especially on the results section).

  1. Fig. 1 is a good photo for demonstrating the outcome of this work. However, the author needs to discuss it in more detail.

Agreed and better described dimensions of the chamber in text (lines 75-82).

  1. Compare the results of this work with other configurations.

It is discussed a comparison with another MSM from a previous report on 4 Rock Sample.

  1. Highlight the new contribution of this work to the research field. I know this work proposes a new reading system and makes the system adaptable to different sensors. Nevertheless, the author should highlight it in Section 3 or Section 4. Some comparison or Schematic of the method may help.

Agreed, we tried to describe it better both in lines 90-94 where we explain why not use other sensors., and lines 377-380 where we compare it with another auxiliary equipment from references [10] and [11]

  1. Considering: remove Fig. 4, font size too small to read, Fig. 5 should retake, rearrange for better quality, Fig. 6, 8, 9 move the labels (a-d) to the right corner, Fig. 7 why two labels (a), (b) are double.

Fig.4 is very important to illustrate the layout of the automation program. It is then very important to explain it, especially by showing its layout, so external users can better grasp the measuring process. All the relevant functions of said program are exposed in Figure (b), (c).

On the other hand, with the exception of Figure 5, all the other ones were agreed and changed. Figure 5 sadly can’t be rearranged without misshaping one of the subfigures

  1. The English, symbols, and presentation need to be improved.

Agreed and Changed

Reviewer 4 Report

Comments and Suggestions for Authors

The Reviewer believes that the manuscript “Multisensor Magnetic Scanning Microscope for Remanent Magnetic Field Measurements” requires the following improvements:

1. In the Introduction, the authors should describe the design of commercially available magnetic scanning microscopes in more detail and discuss the technological challenges of their production. This is necessary to confirm the conclusion about the simplicity of the proposed system.

2. Figure 1 should be supplemented with a scale bar, a general view of the shielded chamber with dimensions.

3. It is necessary to dwell in more detail on the design of the shielded chamber. What alloy is it made of? How much does the surface area of the chamber change when it is opened?

4. Section 2.1.3: it should be indicated what the form of alternating current flowing through the coil is when determining the signal-to-noise ratio.

5. Figure 5: The central part of the holders should be presented in more detail, indicating the dimensions of the recesses in which the test samples were placed.

6. Caption Fig. 8: What is highlighted by the dotted line? (Fig. 8, c and Fig. 8, d).

7. Section 4.3: A more detailed description of the significant difference in magnetization values for the nickel wire sample is required (Table 1). It is worth paying attention to how the wire is placed in the recess of the holder and compare the spatial resolution of the sensors with the characteristic dimensions of the recess.

8. Section 5 should be supplemented with a discussion of the advantages of the developed system.

Author Response

Author's Reply to the Review Report (Reviewer 4)

The Reviewer believes that the manuscript “Multisensor Magnetic Scanning Microscope for Remanent Magnetic Field Measurements” requires the following improvements:

Thank you very much for reviewing our paper

  1. In the Introduction, the authors should describe the design of commercially available magnetic scanning microscopes in more detail and discuss the technological challenges of their production. This is necessary to confirm the conclusion about the simplicity of the proposed system.

On a previous draft of the paper it was mentioned commercially available magnetometers, like VSM, which we used to compare the results we obtained. However, it was decided best to not mention such system because the measurements made for the Ni wire, which was the comparison made, were unreliable (for the same reason probably that we get unreliable measurements for our MSM when changing sensors).

  1. Figure 1 should be supplemented with a scale bar, a general view of the shielded chamber with dimensions.

Corrected. The dimensions of the chamber were explained in a way to save space that would be needed for another figure (lines 72 -77)

  1. It is necessary to dwell in more detail on the design of the shielded chamber. What alloy is it made of? How much does the surface area of the chamber change when it is opened?

Agreed and explained the chamber dimensions (line 72)

  1. Section 2.1.3: it should be indicated what the form of alternating current flowing through the coil is when determining the signal-to-noise ratio.

Agreed and explained that the AC is a square wave current (line 131)

  1. Figure 5: The central part of the holders should be presented in more detail, indicating the dimensions of the recesses in which the test samples were placed.

The dimensions are highlighted (lines 198 -204)

  1. Caption Fig. 8: What is highlighted by the dotted line? (Fig. 8, c and Fig. 8, d).

Agreed and changed.

  1. Section 4.3: A more detailed description of the significant difference in magnetization values for the nickel wire sample is required (Table 1). It is worth paying attention to how the wire is placed in the recess of the holder and compare the spatial resolution of the sensors with the characteristic dimensions of the recess.

Agreed, this section is now explained more, especially on the possible reasons for this difference (lines 360 - 365)

  1. Section 5 should be supplemented with a discussion of the advantages of the developed system.

Advantages have been highlighted on (lines 390-397)

Round 2

Reviewer 1 Report

Comments and Suggestions for Authors

The authors have addressed my concerns. I recommend the revised manuscript for publication.

Reviewer 3 Report

Comments and Suggestions for Authors

Manuscript ID: sensors-2924663

Title: Multisensor Magnetic Scanning Microscope for Remanent Magnetic Field Measurements

Almost all the issues I was concerned about were resolved.

Comments on the Quality of English Language

Minor editing of English language required

Reviewer 4 Report

Comments and Suggestions for Authors

The authors took into account all the reviewer's comments. The manuscript can be accepted for publication.